Fibrochondrogenic potential of synoviocytes from osteoarthritic and normal joints cultured as tensioned bioscaffolds for meniscal tissue engineering in dogs

Warnock Jennifer J. 1 jennifer.warnock@oregonstate.edu
Bobe Gerd 2
Duesterdieck-Zellmer Katja F. 1
1 College of Veterinary Medicine, Oregon State University , Corvallis, OR , United States
2 Linus Pauling Institute, Oregon State University , Corvallis, OR , United States
Arany Praveen
Electronic publication date: 2014 Sep 30
Publication date: 2014
Volume: 2
Electronic Location ID: e581
Received 2014 Jun 10; Accepted 2014 Aug 26
Copyright: © 2014 Warnock et al.
Copyright year: 2014
Copyright holder: Warnock et al.
License: This is an open access article distributed under the terms of the Creative Commons Attribution License, which permits unrestricted use, distribution, reproduction and adaptation in any medium and for any purpose provided that it is properly attributed. For attribution, the original author(s), title, publication source (PeerJ) and either DOI or URL of the article must be cited.
License URL: https://creativecommons.org/licenses/by/4.0/

Keywords: Meniscus, Tissue engineering, Cell culture, Synovium, Osteoarthritis

Funding: Department of Clinical Sciences, College of Veterinary Medicine, Oregon State University This study was funded by the Department of Clinical Sciences, College of Veterinary Medicine, Oregon State University, USA. The funders had no role in study design, data collection and analysis, decision to publish, or preparation of the manuscript. The funders had no role in study design, data collection and analysis, decision to publish, or preparation of the manuscript.

==============================
Meniscal tears are a common cause of stifle lameness in dogs. Use of autologous synoviocytes from the affected stifle is an attractive cell source for tissue engineering replacement fibrocartilage. However, the diseased state of these cells may impede in vitro fibrocartilage formation. Synoviocytes from 12 osteoarthritic (“oaTSB”) and 6 normal joints (“nTSB”) were cultured as tensioned bioscaffolds and compared for their ability to synthesize fibrocartilage sheets. Gene expression of collagens type I and II were higher and expression of interleukin-6 was lower in oaTSB versus nTSB. Compared with nTSB, oaTSB had more glycosaminoglycan and alpha smooth muscle staining and less collagen I and II staining on histologic analysis, whereas collagen and glycosaminoglycan quantities were similar. In conclusion, osteoarthritic joint—origin synoviocytes can produce extracellular matrix components of meniscal fibrocartilage at similar levels to normal joint—origin synoviocytes, which makes them a potential cell source for canine meniscal tissue engineering.

Introduction

Meniscal injury is a common cause of lameness and pain of the dog. Due to the virtually absent healing response in the majority of the meniscus, injured meniscal tissue is commonly removed to relieve the clinical signs of lameness, joint locking, and painful popping. Unfortunately, partial meniscectomy hastens the development of secondary arthritis (Berjon, Munuera & Calvo, 1991; Connor et al., 2009; Cox et al., 1975) and thus patient lameness.

Tissue engineering methods are being investigated to address this challenge of meniscal injury and loss. One of the great obstacles to achieving the reality of tissue engineered menisci is determination of an ideal cell source for in vitro culture and extracellular matrix (ECM) formation. Because cells cannot be synthesized de novo, they must be harvested autologously, or obtained from living or deceased tissue donors. When determining ideal cell sources for tissue engineering, location of the source tissue, quantity of donor tissue available for harvest, and ability to harvest the cells in a minimally invasive fashion must be considered. Autologous cells are particularly attractive because they have a low potential for infectious disease transmission (Pessina et al., 2008) and immunogenic tissue rejection (Hamlet, Liu & Yang, 1997; Ochi et al., 1995; Rodeo et al., 2000).

While producing normal, healthy menisci in vitro is the ultimate goal of tissue engineering, use of normal meniscal cells from a healthy donor site would cause irreversible patient harm and thus is a poor choice for meniscal tissue engineering. Mesenchymal stem cells were recently identified in normal joint—origin canine synovium (Zhang, Dietrich & Lopez, 2013), which could be used towards in vitro meniscal fibrochondrogenesis. When cultured in monolayer, cells obtained from the synovial membrane of normal joints are primarily positive for CD90 (marker for stemness), CD29 (β- integrin), CD44 (hyaluronic acid receptor) and negative for markers of hematopoetic progenitors (CD34) and leukocyte antigens (CD45; Zhang, Dietrich & Lopez, 2013). These cells are also able to undergo chondrogenesis when cultured in pellet form (Zhang, Dietrich & Lopez, 2013). However, clinical use of normal autologous synoviocytes as a tissue engineering cell source would require surgery on another unaltered joint within the patient’s body.

Autologous, osteoarthritic joint—origin synovium has been investigated as a cell source for fibrocartilage tissue engineering in dogs, because of its abundance and ease of harvest during clinically required surgical procedures (Warnock et al., 2012). In vitro, cultured osteoarthritic joint—origin canine synovial membrane cells are plastic adherant and fibroblast—like, and contain populations of cells that can undergo chondrogenesis (Warnock et al., 2011; Warnock et al., 2013), suggesting the presence of mesenchymal stem cells. In vivo synovium also has the ability to form fibrocartilage ECM (Smith et al., 2012; Tienen et al., 2006). Conversely, synoviocytes in osteoarthritic joints secrete a number of inflammatory mediators and destructive matrix metalloproteinases (Benito et al., 2005; Fiorito et al., 2005; Sutton et al., 2007), which could inhibit in vitro fibrochondrogenic potential. For example, canine osteoarthritic joint—origin synoviocytes produce less total collagen than normal joint—origin synoviocytes in monolayer culture (Warnock et al., 2011). This limitation may not be present with improved culture conditions (Warnock et al., 2013). For instance, osteoarthritic joint—origin synovial fluid stem cells require culture as a micro-mass to undergo efficient in vitro chondrogenesis, compared to cells derived from healthy joint fluid (Krawetz et al., 2012).

Thus, the purpose of this study was to evaluate and compare the fibrochondrogenic potential of synoviocytes from osteoarthritic and normal canine joints that were cultured as tensioned bioscaffolds under conditions previously shown to increase meniscal-like ECM content in canine osteoarthritic joint—origin synoviocytes (Warnock et al., 2013). We hypothesized that with the use of this tensioned culture system, there would be no difference in cell viability and fibrocartilage-like ECM formation between tensioned synoviocyte bioscaffolds from normal joints (normal joint—origin tensioned synoviocyte bioscaffolds, “nTSB”) and osteoarthritic joints (osteoarthritic joint—origin tensioned synoviocyte bioscaffolds, “oaTSB”).

Materials and Methods

Tissue harvest

With informed owner consent, synovium was obtained from 12 dogs with naturally occurring (stifle) osteoarthritis as per Institutional Animal Care and Use Committee approval. Dogs were treated for degeneration of the cranial cruciate ligament and medial meniscal injury via exploratory arthroscopy, partial meniscectomy if indicated, and tibial plateau leveling osteotomy. Synovial villi were arthroscopically harvested during routine partial synovectomy using a tissue shaver (Stryker, San Jose, CA) as previously described (Warnock et al., 2012). Synovial villi from the osteoarthritic joints were immediately placed in a 50 ml polypropylene tube containing 40 mL of Dulbeccos’ Modified Eagle’s Media (DMEM, Invitrogen) with 10% fetal bovine serum (FBS, Invitrogen), warmed to 37 °C. The tube was transported immediately to the laboratory and centrifuged at 313 xg, media was decanted, and tissue fragments transferred by pipette and sterile forceps into a digestion solution as described below.

Normal synovium was also harvested from six dogs which were euthanatized via sodium pentobarbital overdose for reasons unrelated to the study, as per the Institutional Animal Care and Use Committee Protocol and in accordance with the American Veterinary Medical Association Humane Euthanasia Guidelines. Dogs were assessed by a Diplomate of the American College of Veterinary Surgeons—Small Animal to not have any orthopedic disease based on medical history, pre-mortem physical examination, and post-mortem gross joint evaluation. Post-mortem, a lateral arthrotomy and patellar tendon transection was performed on each stifle joint. The parapatellar, suprapatellar, lateral, and medial wall synovium were dissected off the joint capsule using a #15 bard parker blade. Synovium was transported as described above. In the laboratory, synovium harvested from normal joints was additionally was minced into 2 × 3 mm pieces using sterile technique.

Cell culture

Osteoarthritic joint—origin synovial villi and normal joint—origin synovium tissue fragments were completely digested with sterile Type 1A clostridial collagenase 10 mg/mL in RPMI 1640 solution (Invitrogen) over 2–6 h at 37 °C. Tissue was deemed to be completely digested when no ECM could be visualized microscopically at 20 × objective magnification. Cells were cultured in monolayer for four passages to isolate Type B fibroblast-like synoviocytes (Vasanjee et al., 2008) and Type C intermediate synoviocytes (Vasanjee et al., 2008) as described previously (Warnock et al., 2012). The following media formulation was used for the duration of culture: high glucose DMEM, supplemented with 17.7% FBS, 0.021 mg/mL glycine, 0.025 mg/mL L-alanine, 0.037 mg/mL L- asparagine, 0.038 mg/mL L-aspartic acid, 0.042 mg/mL L-glutamic acid, 0.033 mg/mL L-proline, 0.030 mg/mL L-serine, 0.23 mg/mL pyruvate, 0.52 mg/mL L-glutamine, 6.75 mg/mL HEPES buffer, 0.15 mg/mL L-ascorbic acid -2 phosphate, 177.0 units/mL penicillin, 177.0 µg/mL streptomycin, and 0.44 µg/mL amphoterocin. The flasks were incubated at 37.6 °C, 5% CO2, 95% humidity, with sterile media change performed every 24 h.

Cell flasks were observed under 10x objective magnification every 24 h to assess confluency. Cells were passaged upon reaching 95% confluence, which was defined as monolayer cell culture with no visible exposed flask surface in between cells, and no overlap of the cells on each other. At harvest and at each passage cell viability counts were performed using the trypan blue exclusion assay (Strober, 2001). At the 4th passage, cells from each joint were transferred into eight 150 cm2 flasks and allowed to become hyperconfluent cell sheets, defined as cells overlapping each other in greater than 100% confluency (Fig. 1). TSB were then made as previously described (Warnock et al., 2013). Briefly, hyperconfluent cell sheets were dislodged off the flask floors (Ando et al., 2008), and each sheet was wrapped over 2.0 cm diameter, 22 ga cerclage wire hoops in three layers, with approximately 0.5 N of tension to avoid tearing, to synthesize TSB. The TSB were placed in 6-well plates in 9.0 mL of the above described culture media, with the free end of the cell sheet facing down to prevent loosening. Bioscaffolds were harvested for analysis after a total of 30 days in culture (Ando et al., 2008; Tan, Zhang & Pei, 2010).

Bioscaffold analyses

Bioscaffold analyses examined presence of ECM components responsible for meniscal form and function. These include type I and type II collagen, (Kambic & McDevitt, 2005; Eyre & Wu, 1983); α- smooth muscle actin (ASM), (Ahluwalia et al., 2001; Kambic, Futani & McDevitt, 2000; Spector, 2001); and glycosaminoglycans (GAG), (Adams & Ho, 1987; Nakano, Dodd & Scott, 1997; Stephan, McLaughlin & Griffith, 1998), including aggrecan (Valiyaveettil, Mort & McDevitt, 2005). Differences in expression of inflammatory mediators or presence of macrophages were investigated as these factors may be associated with decreased in vitro ECM synthesis in osteoarthritic joint—origin synoviocytes (Fiorito et al., 2005; Pei et al., 2008).

Cell viability

One TSB per dog was washed three times in sterile phosphate buffered saline and immersed in 4 µM ethidium homodimer and 6 µM acetomethoxy calcein (calcein –AM) solution (Ethidium homodimer and Calcein AM Live/Dead Viability Assay; Invitrogen, Carlsbad, CA) for 20 min at 37.6 °C, 5% CO2, 95% humidity. Cells were then visualized in at least five regions of the bioscaffolds, (and two in the center and three on the periphery, at approximately the 2, 6, and 10 o’clock positions) using a laser microscope (Eclipse Ti-u Laser Microscope; Nikkon, Japan). The number of viable (green) and non-viable (red) cells per each field counted by hand. Due to the complex three-dimensional nature of the bioscaffolds, these cell counts provided an estimate of cell viability.

Immunohistologic analysis

Two TSB per dog were fixed in 10% buffered formalin, paraffin embedded, and tissue blocks cut in 4 µm sections for histologic and immunohistologic analysis. All slides were labelled with randomly generated acquisition numbers and analyzed in a blinded fashion. Sections were stained with Hematoxylin and Eosin (“H&E”), Masson’s Trichrome, and Toluidine Blue. Cell morphology and general ECM architecture was assessed using H&E; organization and intensity of collagen staining was described using Masson’s Trichrome, and intensity of GAG staining was assessed using Toluidine Blue.

Immunohistochemistry

Immunohistochemistry was performed as previously described (Warnock et al., 2012) for type I collagen (AB749P; 1:100 dilution; Millipore), type II collagen (AB746P; 1:100: Millipore), macrophage MAC387 receptor to determine type A synoviocyte content, (CBL260; 1:200 dilution; Millipore); and alpha smooth muscle actin (M0851; 1:30; Dako). Extracellular and intracellular immunoreactivity intensity and prevalence was scored as previously described (Wakshlag et al., 2011) with some modifications: immunoreactivity was localized to intracellular or extracellular staining, and ECM immunoreactivity intensity was described and scored, as negative (0), mild (1), moderate (2), or strong (3) staining. As determined by hand count, intracellular immunoreactivity and extracellular immunoreactivity was categorized as positive in <10%, 10–50%, or >50% of cells and sample area, respectively. Each of these histologic observations was assigned a score (Table 2). Then a histologic intensity coefficient was calculated for each ECM component, as follows: [[(Extracellular matrix staining intensity score) × (percentage area coverage of positive staining score)] + [(Intracellular staining intensity score) × (percentage positive staining cells score)]]/2 (Table 1).

Table 1 Histologic scoring system.

Extracellular and intracellular immunoreactivity intensity was localized to intracellular or extracellular staining, and ECM immunoreactivity intensity was described and scored as negative (0), mild (1), moderate (2), or strong (3) staining. As determined by hand count, intracellular immunoreactivity and extracellular immunoreactivity was categorized as positive in <10%, 10–50%, or >50% of cells and sample area, respectively. Each of these histologic observations was assigned a score (Table 2). Then a histologic intensity coefficient was calculated for each ECM component, as follows: [[(Extracellular matrix staining intensity score) × (percentage area coverage of positive staining score)] + [(Intracellular staining intensity score) × (percentage positive staining cells score)]]/2 (Table 1).

	None	<10%	10–50%	>50%	None	Mild	Moderate	Strong	
	% Positive staining cells	Intracellular staining intensity	
Intracellular score	0	1	2	3	0	1	2	3	
	% Positive stained extracellular area	Extracellular staining intensity	
Extracellular score	0	1	2	3	0	1	2	3	

Table 2 Gene expression in tensioned synoviocyte bioscaffolds.

The effect of osteroarthritis on fibrochondrogenic gene expression of tensioned synoviocyte bioscaffolds (fold-changes ± SEM). Fold changes were calculated using the following formula: fold change = 2−ΔΔCT = [(CT gene of interest − CT housekeeping gene GAPDH)oaTSB − (CT gene of interest − CT housekeeping gene GAPDH)nTSB].

	Tensioned synoviocyte bioscaffolds (TSB)		
Dog	Normal	Osteoarthritis	SEM	P-value	
Gene:	N = 4	N = 7			
SOX-9	0Reference	+ 1.17	1.54	0.72	
Collagen type I α1	0	+ 6.88	2.62	0.04	
Collagen type II α1	0	+ 71.1	4.48	0.02	
Aggrecan	0	−1.15	1.77	0.84	
Interleukin-6	0	−19.0	2.01	0.001	
Tumor Necrosis Factor α	0	+ 1.49	2.55	0.77	

Tissue weight

One TSB per dog was lyophilized and a dry weight obtained. Samples were digested in 1.0 ml Papain Solution (2 mM Dithiothreitol and 300 µg/ml Papain) at 60 °C in a water bath for 24 h. This papain digest solution was used to obtain double stranded DNA (dsDNA), GAG and collagen content of the bioscaffolds.

DNA quantification

Double stranded DNA quantification assay (The Quant-iT PicoGreenTM Assay; Invitrogen) was performed per manufacturer’s instructions; double stranded DNA extracted from bovine thymus was used to create standards of 1,000, 100, 10, and 1 ng/mL. Standard and sample fluorescence was read by a fluoromoter (Qubit; Invitrogen) at 485 nm excitation/ 528 nm emission, and dsDNA concentration was determined based on the standard curve.

Biochemical ECM analysis

Glycosaminoglycan content was determined by the di-methyl-methylene blue sulfated glycosaminoglycan assay (Farndale, Buttle & Barrett, 1986) using a spectrophotometer (Synergy HT– KC4 Spectrophotometric Plate Reader and FT4software, BioTec, Winooski, VT). Collagen content was determined by Erlich’s hydroxyproline assay, as described by Reddy & Enwemeka (1996). Hydroxyproline content was converted to collagen content using the equation: µg hydroxyproline x dilution factor/0.13 = µg collagen (Ignat’eva et al., 2007), because hydroxyproline consists of approximately 13% of the amino acids in human meniscal collagen (Fithian, Kelly & Mow, 1990). Collagen and GAG content were standardized to tissue dry weight as percentage of dry weight, to compare the experimental neotissues to previously reported normal meniscal ECM content (Eyre & Wu, 1983). Total GAG and collagen content were also reported in µg/neotissue to measure total synthetic activity over the course of 30 days in each TSB. GAG and collagen content were additionally standardized to dsDNA content using the following equations: [µg GAG/µg dsDNA] (Li & Pei, 2011) and [µg collagen/ µg dsDNA] to identify chondrogenic cellular activity of each tested cell origin.

Real-time RT-PCR

One TSB per dog was snap frozen in liquid nitrogen and stored at −80 °C. Total RNA was isolated using the phenol-chloroform extraction (Chomczynski & Sacchi, 1986) with slight modifications. Samples were pulverized using a liquid nitrogen-cooled custom-made stainless steel pulverizer and homogenized in trizol (Trizol; Qiagen Sciences, 0.025 mL/mg of tissue) and mixed with chloroform. The aqueous phase was then treated with isopropanol to precipitate nucleic acids. RNA of samples was purified using on-column DNAse digestion (RNeasy; Qiagen Sciences).

The RNA quality and quantity was determined using capillary electrophoresis (RNA 6000 Nano LabChip Kit, Agilent 2100 Bioanalyzer; Agilent Technologies), and RNA integrity numbers (Imbeaud S, 2005) were determined (2100 Expert software, Agilent Technologies).

First-strand cDNA synthesis was performed from 400 ng total RNA (SuperScript III First-Strand Synthesis System; Invitrogen Life Technologies, Carlsbad, CA) and Oligo-(dT) 20 primers. To control for possible genetic DNA contamination, non reverse-transcribed samples were also processed.

Pre-designed primers and probes (Taq-Man® Primers and Probes; Applied Biosystems Inc., Foster City, CA) were obtained for each of the genes of interest: IL-1β, IL-6, TNF-α, SOX-9 (an embryonic chondrogenic transcription factor), collagen type I α1, collagen type II α1, aggrecan, and the reference gene GAPDH (see Appendix S1). All assays were confirmed to amplify their targets at 95% or greater efficiency using RNA from tissues of interest. Quantitative real-time PCR was performed (StepOnePlus RT-PCR System; Applied Biosystems Inc.) using a proprietary reagent system (TaqMan Gene Expression Master Mix; Applied Biosystems Inc.) Controls included template-free negative controls and non reverse-transcribed negative controls. All samples were run in triplicates and all negative controls were run in duplicates for 40 cycles (15 s at 95 °C, 1 min at 60 °C) after 2 min of incubation with Uracil-DNA Glycosylase at 50 °C, and 10 min at 95 °C of enzyme activation.

Quantitative gene expression was determined in triplicates using the comparative CT method (Schmittgen & Livak, 2008). The gene GAPDH was used as internal control (housekeeping gene). Threshold cycles (CT) for each gene were defined by recording the cycle number at which fluorescence reached a gene-specific threshold. Fold changes for gene expression data were calculated using the following formula: fold change = 2−ΔΔCT = [(CT gene of interest − CT housekeeping gene GAPDH)nTSB − (CT gene of interest − CT housekeeping gene GAPDH)oaTSB].

Statistical methods

A D’Agostino & Pearson omnibus normality test was performed on all data to test for normality. Cell harvest data was non-parametric data and was analyzed with a Wilcoxon matched-pairs signed rank test, and data reported as median and interquartile range. Significance was declared at P < 0.05. Data were analyzed with a statistical software program Graph Pad Prism, San Diego, CA.

The effect of osteoarthritis (osteoarthritic versus normal joint status) on gene expression and ECM composition was analyzed using a 2-tailed Student’s t-test, assuming unequal variances. The effect of osteoarthritis on the histologic scoring of TSB extracellular matrix formation was analyzed using a non-parametric Mann–Whitney U-test; ranking of the histologic scores was performed using a Kruskal-wallis analysis on ranks followed by a Fisher’s exact test. Significance was declared at P < 0.05. Data were analyzed using Statistical Analysis System, version 9.3 (SAS Institute Inc., Cary, NC).

Results

Cell harvest

The mean age of dogs with stifle osteoarthritis was 4.7 years (range: 2–8 years). Breeds represented included: Golden Retriever (1), American Staffordshire Terrier (2), Labrador Retriever (3), Australian Shepherd (1), Rottweiler (2), Boston Bull Terrier (1), Goldendoodle (1), and mixed breed (1), with 7 neutered males, 4 spayed females, and one intact female dog. As observed by a Diplomate of the American College of Veterinary Surgeons–Small Animal, all dogs had marked villous synovial hyperplasia and osteophytosis, and grade 1–2 Outerbridge cartilage lesions of the medial femoral condyle and tibial plateau (Outerbridge, 1961). Cell yield from arthroscopic synovial debris was 1.9 × 106 ± 3.7 × 105 cells per joint, and cells were 99.5% ± 0.002 viable at harvest.

Mean age of dogs with normal stifles was 4.3 years (range: 3–6 years); breeds represented included: Red Tick Hounds (4), Labrador Retriever (1), and American Staffordshire cross (1), with 3 female intact dogs, 2 male intact dogs, and one neutered male. Cell yield per joint was 1.4 × 107 ± 2.6 × 106 per joint and cells were 99.5% ± 0.01 viable. As the entire stifle joint synovial membrane could be harvested post mortem, a greater volume of tissue and thus greater cell numbers were obtained from the normal joints versus arthroscopic harvest of the osteoarthritic joints (P = 0.01).

Cell culture and cell characterization

At 4th passage, cells were transferred into eight 150 cm2 flasks in order to have enough TSB for tissue analyses. This, however, resulted in greater cell seeding numbers for nTSB versus oa TSB. Thus, normal joint—origin synoviocytes were seeded at 1.49 × 107 cells per flask, wherease 6.52 × 106 osteoarthritic joint—origin cells were seeded per flask. At 4th passage, normal joint—origin cells were 99.0 ± 0.4% viable compared with 98.8 ± 0.4% viability of osteoarthritic joint—origin cells (P = 0.85). Culture duration from tissue harvest to hyperconfluent cell membrane formation and synthesis of TSB was 37.6 days and similar for both cell origins (range 20–49 days).

During the first week of tensioned bioscaffold culture, the culture media phenol red pH indicator changed to yellow by the time the 24 h media change was required, indicating marked increase in media acidity. In addition, during the first 7–10 days of culture, approximately 2–3 bioscaffolds per normal and osteoarthritic joint unraveled or slipped off their wire hoops (no group differences observed), and were not analyzed in this study. The typical appearance of intact nTSB and oaTSB is pictured in Fig. 2; thickness of TSB was 2–3 mm.

Figure 1 Hyperconfluent cell sheets.

(A) Representative example (A) of a hyperconfluent cell sheet in a cell culture flask, in monolayer culture, just prior to harvest for formation of a tensioned synoviocyte bioscaffold; (B) phase contrast photomicrograph of the hyperconfluent cell sheet, 10 × objective magnification, bar = 100 µm.

Figure 2 Tensioned synoviocyte bioscaffolds.

Representative samples of a tensioned synoviocyte bioscaffold synthesized from normal joint origin synoviocytes, or nTSB (“N”), and a tensioned synoviocyte bioscaffold from osteoarthritic joint origin synoviocytes, or “oaTSB,” (“OA”).

At harvest, tensioned bioscaffolds from normal dogs had a dry weight of 39.3 mg (range 27.5–50.4 mg), which was more than for oaTSB (23.6 mg, range 10.2–50.1 mg; P = 0.008).

Mean estimated cell viability of nTSB and oaTSB was similar, with 78% of cells viable (range: 72%–86%). Cell viability was not associated with peripheral versus central location on the TSB. Laser microscopy revealed cells with fusiform, fibroblastic morphology, oriented parallel with the vector of tension, as well as the presence of acellular, circular regions in the bioscaffolds. Hematoxylin and eosin staining revealed highly cellular bisocaffolds, with layers of fibroblastic cells organized in parallel, as sheets or bands, or variably arranged in whorls, with eosinophilic ECM (Fig. 3). Subjectively, both nTSB and oaTSB had heterogeneous extracellular matrix architecture and cell distribution, with regions of dense cellularity and regions of dense extracellular matrix (Fig. 3).

Figure 3 Bioscaffold cellularity and alpha-smooth muscle actin expression.

Hematoxylin and Eosin stain of normal joint—origin synoviocyte bioscaffolds (“N H&E”) and osteoarthritic joint—origin synoviocyte bioscaffolds (“OA H&E”). Immunohistochemisty for alpha smooth muscle actin (ASM) of normal joint—origin synoviocyte bioscaffolds (“N ASM”) and osteoarthritic joint—origin synoviocyte bioscaffolds (“OA ASM”). Note the more extensive expression of ASM in the osteoarthritic joint—origin synoviocyte bioscaffold cells, and regional strong ASM expression along the periphery of spontaneously forming defects (*) in the normal and osteoarthritic joint—origin bioscaffolds. Immunohistochemistry negative controls of normal joint—origin synoviocyte bioscaffolds (“N NC”) and osteoarthritic joint—origin synoviocyte bioscaffolds (“OA NC”). 10 × objective magnification, bar = 100 µ m.

Percent dsDNA content was used to quantify tissue cellularity. Despite an initial higher seeding cell count at 4th passage, dsDNA accounted for 0.11 ± .02% of nTSB dry weight, versus 0.21 ± 0.03% of oaTSB dry weight (P = 0.01).

Immunohistologically, oaTSB had more ASM positive cells than nTSB; the median histologic score for nTSB was 6 versus 9 for oaTSB (p = 0.0102, Fig. 4). Nine of 12 oaTSB had the highest possible ASM histologic scores of 9, whereas none of the nTSB achieved a perfect score of 9 (P = 0.009, Fig. 4). In 50% of all bioscaffolds, ASM positive cells were concentrated around the bisocaffold periphery and around the margins of what appeared to be spontaneously forming bioscaffold defects ranging from 70–600 µm (Fig. 3). These defects corresponded with the circular acellular regions viewed on laser microscopy. The other 50% of bioscaffolds did not contain circular defects, nor did ASM expression seem to be geographically localizable.

Figure 4 Histology scores for tensioned synoviocyte bioscaffolds.

Scatter plots for histology scores for type I collagen, type II collagen, glycosaminoglycan, and alpha-smooth muscle actin in normal joint—origin tensioned synoviocyte bioscaffolds versus osteoarthritic joint—origin tensioned synoviocyte bioscaffolds, showing the median (thin line) and interquartile range (bolded bars). Histologic scores for collagens type 1 and 2 were calculated as follows: Histologic score= [[(% positive staining cells × intracellular staining intensity) + (% positive stained extracellular area × extracellular staining intensity)]/2]. The histologic score for alpha smooth muscle actin (ASMA) was calculated by (% positive staining cells × intracellular staining intensity). The histologic score for glycosaminoglycan (GAG) was calculated by (% positive stained extracellular area × extracellular staining intensity). An (∗) denotes statistical significance.

Gene expression

The oaTSB had a greater gene expression of type I collagen (7-fold increase; P = 0.04) and type II collagen (71-fold increase; P = 0.02) and a lower gene expression of interleukin-6 (19-fold decrease; P = 0.001) versus nTSB. No significant changes were observed for relative expression of SOX-9 (P = 0.72), aggrecan (P = 0.84), and tumor necrosis factor-α (P = 0.77; Table 2). Interleukin-1β was not expressed at detectable levels in any bisocaffolds.

Glycosaminoglycan content

The total GAG content of oaTSB was lower than the GAG content of nTSB (P = 0.02; Table 3). After adjustment for dry weight or DNA content, no significant group differences were observed.

Table 3 Extracellular matrix and dsDNA content of tensioned synoviocyte bioscaffolds.

The effect of osteoarthritis on extracellular matrix and double stranded- DNA composition of tensioned synoviocyte bioscaffolds. Data is reported as mean ± SEM.

	Tensioned synoviocyte bioscaffolds (TSB)		
Dog	Normal	Osteoarthritis	P-value	
	N = 6	N = 12		
Concentrations (µ g/neotissue):				
Glycosaminoglycan	684 ± 74	434 ± 44	0.02	
Collagen	4855 ± 1270	3302 ± 392	0.29	
DNA	47.4 ± 11.9	42.3 ± 5.2	0.71	
Proportion (% dry weight):				
Glycosaminoglycan	1.73 ± 0.11	2.05 ± 0.16	0.11	
Collagen	12.1 ± 2.4	16.6 ± 2.4	0.20	
DNA	0.111 ± 0.020	0.214 ± 0.030	0.01	
Index (µ g/µ g dsDNA):				
GAG	18.7 ± 3.9	11.4 ± 1.6	0.13	
Collagen	132 ± 36	92.0 ± 17.5	0.35	

Glycosaminoglycan was deposited regionally in all bioscaffolds but more GAG staining was observed in oaTSB than in nTSB. Median GAG histologic score was 1.0 for nTSB and 3.0 for oaTSB (P = 0.0007, Figs. 4 and 5). Only 1 of 6 nTSB had a GAG histologic score above 1, whereas 11 of 12 oaTSB had GAG histologic score above 1 (P = 0.004, Fig. 4).

Figure 5 Histologic analysis of glycosaminoglycan content of tensioned synoviocyte bioscaffolds.

Toluidine Blue staining for glycosaminoglycan of normal joint—origin tensioned synoviocyte bioscaffolds (“NTB”) and osteoarthritic joint—origin tensioned synoviocyte bioscaffolds (“OATB”). 10 × objective magnification, bar = 100 µm.

Collagen content

There was no difference in quantified total collagen content of oaTSB and nTSB (Table 3). Similar results were observed after adjustment for dry weight or DNA content.

Masson’s Trichrome staining revealed collagen deposited in bands, sheets, and whorls, containing and surrounded by numerous fibroblastic cells lined in parallel with the orientation of the collagen (Fig. 6). A significant difference in the median type I collagen histologic scores of nTSB and oaTSB could not be detected, which were 7.5 and 6.0, respectively (P = 0.11, Fig. 4). However, 4 of 6 nTSB had a type I collagen histologic score greater than 7.5, vs. only 1 of 12 oaTSB had a score of 7.5 (P = 0.02, Fig. 4). Histologically, nTSB had more type II collagen than oaTSB (Fig. 6); median type II collagen histologic scores were 4.0 in nTSB and 2.5 in oaTSB, (P = 0.03, Fig. 4). None of the oaTSB had a score greater than 2.5 whereas 5 of 6 nTSB had a collagen type II histology score of 2.75 (P = 0.0007, Fig. 4).

Figure 6 Histologic analysis of collagen content of tensioned synoviocyte bioscaffolds.

Masson’s Trichrome staining for collagen of normal joint—origin synoviocyte bioscaffolds (“NMT”) and osteoarthritic joint—origin synoviocyte bioscaffolds (“OAMT”). Immunohistochemistry for type I collagen and type II collagen of normal joint—origin synoviocyte bioscaffolds (“NCOL1” and “NCOL2”) and osteoarthritic joint—origin synoviocyte bioscaffolds (“OACOL1” and “OACOL2”). In this example the type I collagen ECM of both bisocaffolds is moderately positive. For type II collagen, the cells are moderately immunoreactive and the ECM is mildly immunoreactive in the normal joint—origin synoviocyte bioscaffold, while the cells and ECM of the osteoarthritic joint—origin bioscaffold are mildly immunoreactive. 10 × objective magnification, bar = 100 µm.

Synovial macrophage content

Based on immunohistochemistry, no macrophages (Type A synoviocytes) were found in any bisocaffolds (Fig. 7).

Figure 7 Immunohistochemical analysis for macrophages.

Immunohistochemistry for the macrophage MAC387 receptor, in a lymph node (positive control, “PCMAC”), a normal joint—origin bioscaffold (“NMAC”) and an osteoarthritic joint—origin synoviocyte bioscaffold (“OAMAC”). Negative controls are as pictured in Fig. 3. Note the lack of immunoreactivity in the bioscaffold samples. The positive control, a canine lymph node, is shown at 20 × objective magnification, bar = 100 µm, to allow clearer viewing of the macrophages. To show adequate representation of the bioscaffold neotissues, NMAC and OAMAC are pictured at 10 × objective magnification, bar = 100 µ m.

Discussion

Previous studies comparing in vitro canine synoviocyte fibrochondrogenesis in monolayer culture (Warnock et al., 2011), and canine synoviocyte chondrogenesis in micromass culture (Krawetz et al., 2012) concluded that osteoarthritic synoviocytes had inferior in vitro fibrochondrogenic potential, compared with normal synoviocytes. Fiorito and colleagues (2005) came to a similar conclusion in a study comparing in vitro chondrogenesis of human synoviocytes grown in pellet culture, as determined by histologic analysis. In contrast, with the culture conditions in the present study, especially providing conditions for self-tensioning, cells originating from osteoarthritic joints increased type I and II collagen gene expression, and oaTSB contained similar total collagen content, as compared to nTSB. While tissue dry weight and thus total GAG content of oaTSB was lower than nTSB, a significant difference in GAG content standardized to dry weight and cellularity could not be detected between oaTSB and nTSB. Histologic analysis using toluidine blue, a semi-quantitative measure of GAG, revealed more GAG deposition in oaTSB than nTSB. Thus, the greater dry weight of nTSB versus oaTSB was likely due to unmeasured ECM components such as fibronectin, type III and VI collagen, and vitronectin (Ando et al., 2007; Ando et al., 2008), which are found in native synovium (Okada et al., 1990; Price, Levick & Mason, 1996). These findings also indicate that given the chance to self-tension, autologous, diseased synoviocytes can produce the ECM components of fibrocartilage in vitro at a comparable level of normal joint—origin synoviocytes.

The unstable mechanical environment and inflammatory environment of the cranial cruciate ligament deficient joint favors synovial intimal hyperplasia and synovial membrane and joint capsule fibrosis (Bleedorn et al., 2011; Buckwalter, 2000; Oehler et al., 2002; Smith et al., 1997), all of which were encountered in the osteoarthritic joints in the present study. The in vivo pathogenic synovial hyperplasia may have accounted for the collagen gene upregulation seen in oaTSB. Rat and human osteoarthritic synoviocytes spontaneously express TGFβ-1 and its receptor (Fiorito et al., 2005; Mussener et al., 1997), which is a pro-collagen and chondrogenic growth factor (Daireaux et al., 1990; Leask & Abraham, 2004; Miyamoto et al., 2007; Pangborn & Athanasiou, 2005a; Pangborn & Athanasiou, 2005b; Pei, He & Vunjak-Novakovic, 2008). Upregulation of TGFβ-1 and its receptor may also be a plausible mechanism for oaTSB collagen gene upregulation. Collagen II upregulation seemed to occur independently of SOX-9 expression, a finding duplicated in cultured human osteoarthritic chondrocytes (Aigner et al., 2003). Additionally, decreased expression of IL-6 gene may be a mechanism for the observed upregulation of type II collagen genes in oaTSB; IL-6 has been found to inhibit chondrogenic differentiation of murine marrow mesenchymal cells (Wei et al., 2013). Further research is required to confirm the mechanism of hyaline chondrogenic ECM formation in canine TSB, through immunohistochemistry of TGFbeta-receptor and SMAD-family protein expression (Xu et al., 2012).

Despite equal quantities of non-specific collagen in nTSB and oaTSB, immunohistologic analysis revealed less type I and type II collagen in oaTSB, particularly in the ECM. Post translation regulation by prolyl-4-hydroxylases (Grimmer et al., 2006) or ECM degradation by synovial matrix metalloproteinases (Fiorito et al., 2005) may have decreased oaTSB accumulation of type I and II collagen accumulation, despite increased collagen gene expression. IL-6 has also been found to increase gingival fibroblast synthesis of type I collagen in vitro (Martelli-Junior et al., 2003), and increase type I collagen synthesis by tenocytes in vivo (Andersen et al., 2011). It is possible that the decreased IL-6 gene expression in oaTSB synoviocytes also decreased type I collagen formation as seen on histologic analysis. One weakness of our study was that expression of type I and II collagen was not corroborated with a Western blot, nor quantified via ELISA, to further our understanding of this discrepancy between histologic collagen expression and collagen gene expression. Additionally we did not characterize the percentage and type of mesenchymal progenitor cells present in normal versus osteoarthritic synovium; difference in number and chondrogenic potential of these cells may have also accounted for a difference in collagen ECM formation.

Other osteoarthritic cell types, such as chondrocytes, have reduced cell proliferation compared to normal cells in monolayer culture (Acosta et al., 2006). In contrast, oaTSB contained more dsDNA per dry weight than nTSB, despite the lower harvest cell yield and lower cell seeding density at 4th passage of osteoarthritic joint—origin synoviocytes. There was an intrinsic weakness of our study; by clinical necessity, synovium from osteoarthritic joints was harvested using a different technique (arthroscopy) than the normal joints (arthrotomy), and more synoviocytes can be obtained via arthrotomy. Although cell growth kinetics was not the focus of this study, cell culture media containing 17.7% FBS likely provided mitotic stimuli to support and increase oaTSB cellular proliferation. The markedly hyperplastic state of the synovium in vivo may also have primed the osteoarthritic cells to continue to proliferate in vitro. Cell viability was high at harvest and at the start of 4th passage, but declined in all TSB possibly due to the long culture period. Additionally, as evidenced by media color changes, inadequate nutrient delivery to TSB in the culture wells and daily shifts in pH may have also led to nTSB and oaTSB cell mortality. This cell mortality may have affected ECM formation in both groups: the collagen content of nTSB (12%) and oaTSB (16%) did not reach that of the healthy meniscus, at 60–70% of dry weight (McDevitt & Webber, 1990), although the GAG content of nTSB (1.7%) and oaTSB (2%) did approximate the 2–3% GAG per dry weight of the whole meniscus (McDevitt & Webber, 1990; Stephan, McLaughlin & Griffith, 1998).

Consistent with prior studies (Warnock et al., 2013), all oaTSB and nTSB in the present study were negative for any macrophages, which have been reported to contaminate human osteoarthritic synoviocyte monolayer cultures and reduce in vitro chondrogenic activity (Pei et al., 2008) by contributing to the inflammatory milieu. In the present study, 4 passages and long term culture as TSB likely eliminated any non-adherent cells, including synovial macrophages (Krey, Scheinberg & Cohen, 1976). Synovium from osteoarthritic joints has also been found to express inflammatory cytokines (Fiorito et al., 2005). Both nTSB and oaTSB expressed similar RNA quantity of the TNFα gene, indicating an inflammatory response in in vitro culture (Lindroos et al., 2010), independent of the diseased status of the cell origin. Paradoxically, IL-6 expression was decreased in oaTSB. Although the exact reason for this is unclear, decreased IL-6 gene expression may represent the response of synoviocytes from osteoarthritic joints to the change in environment; from the high motion, inflamed stifle containing multiple injured cell types (ligament, cartilage, meniscus, synovium) to the static tension of TSB culture and high FBS concentration cell culture media.

Decreased IL-6 in oaTSB may have reflected better mechanical homeostasis (Asparuhova, Gelman & Chiquet, 2009; Chan et al., 2011; Gardner et al., 2012) in the cells in oaTSB: the majority of cells in oaTSB were uniformly positive for ASM, while 10–50% of nTSB cells were ASM positive. Synoviocytes increase expression of intracellular ASM in response to TGFβ-1 (Xu et al., 2012). Endogenous receptivity in osteoarthritic origin-joint synoviocytes to TGFβ-1 present in FBS (Goddard, Grossman & Moore, 1990; Mussener et al., 1997) may explain increased ASM in the cells of oaTSB. In the present study, staining for ASM was positively associated with the formation of circular defects, indicating that the ECM was not strong enough to prevent tears from forming during ASM-mediated self-tensioning (Kambic, Futani & McDevitt, 2000; Vickers et al., 2004; Warnock et al., 2013). Given the higher dsDNA content of oaTSB and the high cellularity of the TSB, these defects may have also been caused by increased cell turnover.

Conclusion

When cultured as TSB in high concentrations of FBS, osteoarthritic joint—origin synoviocytes can produce ECM components of meniscal fibrocartilage at similar levels to normal joint—origin synoviocytes. Potential reasons for this include increased collagen and decreased IL-6 gene expression and the greater GAG and ASM staining in oaTSB compared with nTSB Osteoarthritic joint—origin synoviocytes are a viable cell source toward meniscal tissue engineering. Further investigation of culture environments to optimize cell viability and ECM formation and strength are justified due to the promising data reported here.

Supplemental Information

Supplemental Information 1 Supporting data file for open data policy

Data sheet showing cell viability, ECM content, gene expression, and histologic scores for tensioned bioscaffolds originating from cell harvested from normal and osteoarthritic joints.

Click here for additional data file.

Supplemental Information 2 Informed consent form for animal owners

Click here for additional data file.

Supplemental Information 3 Animal care and use exemption form

Click here for additional data file.

Appendix S1 Appendix 1

Assays used for RT-PCR

Click here for additional data file.

The authors give profound thanks to Jesse Ott, for technical assistance and expertise in performing some of the assays used in this study.

Additional Information and Declarations

Competing Interests

Author Contributions

The authors of this manuscript have no financial, professional, personal, or other competing interests, which would have otherwise caused bias, to declare regarding this work.

Jennifer J. Warnock conceived and designed the experiments, performed the experiments, analyzed the data, contributed reagents/materials/analysis tools, wrote the paper, prepared figures and/or tables, reviewed drafts of the paper.

Gerd Bobe analyzed the data, reviewed drafts of the paper.

Katja F. Duesterdieck-Zellmer performed the experiments, analyzed the data, contributed reagents/materials/analysis tools, reviewed drafts of the paper.

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
