# Peer review of "Fibrochondrogenic potential of synoviocytes from osteoarthritic and normal joints cultured as tensioned bioscaffolds for meniscal tissue engineering in dogs"

_PeerJ, doi:10.7717/peerj.581_

## Round 0.1 · original submission · Minor Revisions

· Academic Editor

Minor Revisions

Overall, this work is well written and appropriate for this journal audience. Kindly address the minor concerns raised by the reviewers.

·

Basic reporting

The submission is written clearly and conforms to the publication standards. Authors demonstrate a sufficient knowledge of the subject, figure data is relevant to the article contents.

Experimental design

Experimental designs are well described, with a sufficient number of repeats and necessary controls. Methods are well described.

Validity of the findings

The main point of the article is that both normal and osteoarthritic cells are capable of producing a sufficient levels of collagen, regardless of the cell origin (either from the normal or arthritic joint). This is an interesting finding and indeed is a valid point to study.
There are several key aspects that need to be discussed and presented further, prior to the accepting the publication:
1.The obvious discrepancy between the levels of collagen gene expression and IHC findings needs to be firmly established by an additional method (e.g. Western blot, immunoprecipation or collagen-type ELISA)
2. Suggested signaling mechanisms (e.g. TGF-beta1) should be confirmed by IHC for TGFbeta-receptor and SMAD-family proteins

Additional comments

It is a good study, with obvious translational and basic science importance- your conclusions will bear more validity, once you'll be able to provide further evidence for your findings.

Reviewer 2 ·

Basic reporting

No Comments

Experimental design

I only have one comment, regarding initial characterization of the cells, see below.

Validity of the findings

I only have one comment, regarding initial characterization of the cells, see below.

Additional comments

Overall this is a nicely done study detailing the effects of a 3D culture environment potentially overcoming an inherent limitation that source cells have in being suitable for donors as tissue grafts. I do have one major issue with this study that I detail below. The authors need to address this, possibly by doing more characterization of the initial cell population they derive. There are also some minor grammar issues (for example: line 112 should read “Bioscaffolds analyses were performed to assess the presence…”).

Major:
There is one major issue with this study. Given that the conclusion is that cells from “normal” dogs have the same ability to synthesize ECM, and similar levels of IL-6, to the cells derived from dogs with synovial hyperplasia, it is odd that the authors used different methods to isolate the cells. I understand that these experiments are difficult and the animals used are not like rodents (e.g. the authors cannot redo experiments), but it would make the work much stronger if the authors could provide some biomarkers or other assessments (FACs, ICC, etc.) to prove that they have the same starting population of cells, going into the tissue constructs. The first author has other manuscripts, referenced here, on these cells, so at the very least, she can provide a sentence or two indicating what this initial characterization showed, if she did that in those previous papers.
Minor:
Materials and methods: For reproducibility, the authors should supply a micrograph that details exactly what they mean by 95% confluency, as different researchers define this confluency point differently. Image of “hyperconfluent” cells also should be provided.
Cell viability: can the authors provide at least a rough estimate of scaffold thickness, and the viability of cells in the center?
Histologic analysis: How thick were the paraffin sections? Was the immunohistochemistry analyzed in a blinded manner?
Data: In addition to representative images, it would be nice to have quantitative analysis with error bars. Readers should get an idea of the inherent variability of these assays.

---

## Round 0.2 · Minor Revisions

· Academic Editor

Minor Revisions

I am happy to inform you that your manuscript is provisionally accepted. Please include the requested figure and comment from reviewers in your final draft at your earliest convenience, the manuscript will not be re-reviewed.

·

Basic reporting

Article is well written and conforms to PeerJ policies

Experimental design

Studies are well described.

Validity of the findings

Findings are validate by an appropriate analysis. Authors also have incorporated an important limitations of their findings in discussion

Additional comments

1. "Based on immunohistochemistry, no macrophages were found in any bisocaffolds. "-please show relevant image with negative/positive controls.

Reviewer 2 ·

Basic reporting

The authors have answered all the questions we raised during the review, and provided new data to verify the significance of the differences they saw with ICC. Although it would be better for their cells from different animals to be isolated with the same procedure, I understand, based on their explanation and edited MS, why the authors didn't do this.

My only comment now is, in Figure 5, can the authors add a sentence or two in the text mentioning why the OATB and NTB H&E staining looks so different? It appears that cells in the OATB are better aligned, and that the distribution of cells is more heterogenous than in NTB tissue. Since this is a very minor change I don't need to see the new text.

Experimental design

No Comments

Validity of the findings

No Comments

Additional comments

Please see above

---

## Round 0.3 · accepted · Accept

· Academic Editor

Accept

The reviewer comments have been appropraitely addressed.